# Enhancing Instance Segmentation in Agriculture: An Optimized YOLOv8 Solution

**DOI:** 10.3390/s25175506

**Published:** 2025-09-04

**Authors:** Qiaolong Wang, Dongshun Chen, Wenfei Feng, Liang Sun, Gaohong Yu

**Affiliations:** 1School of Mechanical Engineering, Zhejiang Sci.-Tech University, Hangzhou 310018, China; 2023220503143@mails.zstu.edu.cn (Q.W.); 2023010501002@mails.zstu.edu.cn (D.C.); 2024210501013@mails.zstu.edu.cn (W.F.); yugh@zstu.edu.cn (G.Y.); 2Key Laboratory of Agricultural Equipment for Hilly and Mountainous Areas in Southeastern China (Co-Construction by Ministry and Province), Ministry of Agriculture and Rural Affairs, Hangzhou 310018, China; 3Zhejiang Provincial Key Laboratory of Agricultural Intelligent Sensing and Robotics, Hangzhou 310018, China

**Keywords:** YOLOv8n-seg, instance segmentation, agricultural scenes, CPCA attention mechanism, RFEM

## Abstract

To address the limitations of traditional segmentation algorithms in processing complex agricultural scenes, this paper proposes an improved YOLOv8n-seg model. Building upon the original three detection layers, we introduce a dedicated layer for small object detection, which significantly enhances the detection accuracy of small targets (e.g., people) after processing images through fourfold downsampling. In the neck network, we replace the C2f module with our proposed C2f_CPCA module, which incorporates a channel prior attention mechanism (CPCA). This mechanism dynamically adjusts attention weights across channels and spatial dimensions to effectively capture relationships between different spatial scales, thereby improving feature extraction and recognition capabilities while maintaining low computational complexity. Finally, we propose a C3RFEM module based on the RFEM architecture and integrate it into the main network. This module combines dilated convolutions and weighted layers to enhance feature extraction capabilities across different receptive field ranges. Experimental results demonstrated that the improved model achieved 1.4% and 4.0% increases in precision and recall rates on private datasets, respectively, with mAP@0.5 and mAP@0.5:0.95 metrics improved by 3.0% and 3.5%, respectively. In comparative evaluations with instance segmentation algorithms such as the YOLOv5 series, YOLOv7, YOLOv8n, YOLOv9t, YOLOv10n, YOLOv10s, Mask R-CNN, and Mask2Former, our model achieved an optimal balance between computational efficiency and detection performance. This demonstrates its potential for the research and development of small intelligent precision operation technology and equipment.

## 1. Introduction

With the advancement of global agricultural automation, there is growing demand for efficient and precise crop management. The agricultural sector’s increasing reliance on automated technologies has led to the rapid adoption of smart equipment in field operations and harvesting processes [1,2]. However, traditional manual methods suffer from inefficiency, high costs, and significant errors, making the development of automated farm machinery an inevitable trend. Yet current agricultural machinery often lacks sufficient environmental perception capabilities, particularly when encountering obstacles or irregular terrains in complex farmland environments, which may result in operational errors or reduced efficiency [3]. Therefore, enhancing agricultural machinery’s environmental awareness and improving its object segmentation capabilities in complex scenarios represent crucial steps toward achieving agricultural automation. For instance, recent studies have focused on the development of vision-based environmental perception systems that leverage deep learning technologies to identify and segment various objects within agricultural settings, including crops, obstacles, and field pathways [4,5].

To improve the obstacle avoidance capabilities of agricultural machinery, environmental perception technology has emerged as a significant area of research within the domain of agricultural autonomous navigation. Presently, this technology predominantly employs a combination of light detection and ranging (LiDAR), satellite remote sensing systems, and machine vision [6]. For instance, Kragh et al. introduced a method for 3D point cloud object detection and terrain classification utilizing a single multi-beam LiDAR scan, aimed at addressing the challenges faced by autonomous agricultural vehicles operating in unstructured environments. This approach employs a support vector machine (SVM) classifier to categorize the 3D points into three categories: ground, vegetation, and objects. Experimental findings indicated that this method achieved an overall classification accuracy of 91.6% in actual farm settings, with an object detection accuracy of 81.1% and a vegetation classification accuracy of 97.5% [7].

The rapid advancement of deep learning technologies has led to the increasing application of deep learning-based environmental perception methods in instance segmentation tasks within agricultural contexts owing to their notable robustness and generalization capabilities. For instance, researchers have explored the potential of utilizing autonomous robots as alternatives to herbicides for precise weeding. They developed and assessed an instance segmentation convolutional neural network specifically designed to segment and identify individual plants in images captured by agricultural robots. By fine-tuning the hyperparameters of the Mask R-CNN model, the study yielded promising detection outcomes, successfully eliminating between 10% and 60% of weeds without misidentifying crops. Furthermore, instance segmentation facilitates the precise localization of weeding points, thereby opening new avenues for optimizing weeding operations [8]. In the domain of plant disease detection, a deep learning-based system was created to identify diseases in tomato plants. This system was trained using an Inception Net model on a dataset comprising 18,161 images of tomato leaves, resulting in high accuracy in disease identification. The enhanced U-Net model demonstrated superior performance in segmentation tasks, achieving an accuracy rate of 98.66%. InceptionNet also excelled in classification tasks, attaining a binary classification accuracy of 99.95%, thereby surpassing existing technologies in the realm of plant disease detection [9]. In research on fruit detection based on instance segmentation, an improved method for automatic segmentation and recognition of cucumbers was proposed [10], combining YOLO-v3, SSD, U-Net, and transfer learning. Researchers used an improved U-Net for pixel-level cucumber segmentation, followed by recognition using YOLO-v3 or SSD. Results showed that YOLO-v3 outperformed SSD in object recognition, with an F1 score exceeding 0.95 and an average precision (AP) as high as 99%. The mIOU value and average pixel accuracy of the U-Net model were 94.24% and 97.46%, respectively. The combined U-Net and YOLO-v3 model performed best in predictions, with an AP value increase of 6%. This method improved the accuracy of cucumber detection, aiding in the enhancement of positioning and grasping for harvesting robots. The authors of ref. [11] proposed an improved Mask R-CNN method for detecting cucumber fruits. This method optimized the region proposal network (RPN) and introduced a logical green (LG) operator to effectively filter out background elements, while also adjusting the anchor box parameters. The enhanced Mask R-CNN achieved an F1 score of 89.47% during testing, with a processing time of 0.3461 s, and demonstrated lower positional deviation compared to alternative methods, thereby exhibiting exceptional performance.

Although numerous studies have investigated the utilization of deep learning-based RGB cameras within agricultural contexts, the majority predominantly concentrate on object detection and classification, with relatively limited research dedicated to instance segmentation. Current instance segmentation models encounter a variety of challenges in agricultural settings, including variations in scenes, fluctuating lighting conditions, and resemblance between target objects and their backgrounds. In intricate agricultural environments, the morphological and chromatic distinctions among crops, weeds, soil, tools, and other components are often minimal, thereby complicating the application of conventional segmentation techniques. In terms of small object detection capability, existing models (such as Mask R-CNN and the YOLO series) generally exhibit low recognition accuracy for small objects (e.g., distant people and small agricultural machinery) in multi-scale object detection, making them prone to missed detections or incorrect segmentation. When dealing with complex scenes, due to the presence of numerous visual interferences in farmland environments (such as similar colors between crops and soil, drastic lighting changes, and severe occlusions), current methods have not fully optimized feature extraction and attention mechanisms, resulting in blurred segmentation boundaries and category confusion. Although lightweight models like YOLOv8 have advantages in speed, research on further reducing parameter count and computational complexity while maintaining high accuracy remains limited, restricting their deployment on resource-constrained embedded devices. Additionally, most existing methods are trained on general datasets and have limited modeling capabilities for agriculture-specific categories (such as field ridges and lodged areas) and structural features (such as irregular terrain and crop rows), leading to poor generalization performance.

Consequently, this study seeks to develop an instance segmentation methodology tailored for complex agricultural scenes, utilizing the YOLOv8-seg model, with the objective of improving the model’s segmentation accuracy and stability under these demanding conditions. The primary contributions of this research include:(1)A small target detection layer is added, which specifically processes feature maps that have been downsampled by a factor of four from the original image, thereby significantly improving the detection accuracy for small targets (such as people).(2)The C2f_CPCA module is proposed, which introduces a channel priority convolutional attention (CPCA) mechanism in the C2f_ module. This mechanism effectively captures relationships at different spatial scales by dynamically adjusting the attention weights in both channel and spatial dimensions. Therefore, the feature extraction and category recognition ability of agricultural specific scenarios are improved, and the robustness of the model in complex background is enhanced.(3)A C3RFEM module is proposed, which combines dilated convolution and a weighted layer, adding it to the backbone network to enable the model to extract richer features across different receptive field ranges and further improve the generalization ability.(4)The calculation efficiency of the model is further optimized to achieve a better balance between accuracy and speed under the premise of maintaining high segmentation accuracy.

## 2. Materials and Methods

### 2.1. YOLOv8-Seg Instance Segmentation Network Structure

YOLOv8 represents a sophisticated model developed by the Ultralytics team (5001 Judicial Way Frederick, Ballenger Creek, MD, USA) and various contributors, building upon the YOLOv5 framework to facilitate real-time multitasking capabilities. This model demonstrates exceptional performance on edge devices and in industrial applications, characterized by its remarkable versatility. The fundamental architecture of YOLOv8 is comprised of three primary components: the backbone, the neck, and the head.

Drawing inspiration from the YOLOv7 ELAN architecture, the backbone of YOLOv8 employs the C2f module as its foundational convolution unit. This module effectively enhances the gradient information flow by integrating two parallel branches for gradient propagation. Furthermore, the model incorporates spatial pyramid pooling fusion (SPPF) technology. SPPF stands for spatial pyramid pooling fast. It is a computationally optimized version of the original SPP module, first proposed by Ultralytics in YOLOv5 and subsequently adopted in YOLOv8. It serves as an efficient and powerful multi-scale feature extractor, playing a crucial role in the exceptional performance of YOLOv5/v8/v10 series models, which significantly bolsters the model’s generalization capabilities by extracting contextual information from images across various scales. In terms of the neck structure, YOLOv8 departs from the conventional convolutional framework during the upsampling phase, opting instead to replace the C3 module with the C2f module to optimize overall performance. The output head has been substantially refined compared to YOLOv5, with the classification and detection heads for detection tasks being substituted with a decoupled-head structure [12], thereby facilitating the transition from anchor-based to anchor-free methodologies. For segmentation tasks, YOLOv8 leverages the prototype generation concept from YOLACT, achieving segmentation objectives through a linear combination of prototype masks and instance mask coefficients.

In the context of sample matching strategies, YOLOv8 moves away from the previously utilized intersection over union (IoU) matching and unilateral ratio allocation techniques, opting instead for the task-aligned assigner method [13]. This approach prioritizes samples by ranking them according to the weighted values of classification and regression scores, thereby facilitating the selection of Top K positive samples. For computation of the loss function, the classification branch continues to employ binary cross-entropy loss (BCE Loss), while the regression branch incorporates the distribution focal loss (DFL) function [14] to mitigate the challenges associated with class imbalance. The innovation of the DFL loss function is characterized by its transformation of bounding box regression values from a deterministic distribution to a probabilistic distribution, which enhances the computation of focal loss and subsequently improves the network’s focus capability.

YOLOv8 not only provides a framework for model training but also performs basic tasks such as object detection, instance segmentation, image classification, and pose estimation. It is divided into five types based on different architecture scales, n, s, m, l, and x, with the number of model parameters positively correlated with accuracy. Among them, the structure of the YOLOv8-seg model is shown in Figure 1.

### 2.2. Improvement of the YOLOv8-Seg Instance Segmentation Network Structure

The YOLOv8n-seg model, characterized by its minimal parameter count and computational demands, was selected as the foundational model for instance segmentation. This study proposes an instance segmentation algorithm tailored for rice harvesting environments, based on the enhanced YOLOv8-seg model, as illustrated in Figure 2. In contrast to the original YOLOv8-seg model, the proposed algorithm incorporates an additional detection layer specifically designed for small targets, supplementing the existing three detection layers. Furthermore, the C2f module within the neck network is substituted with the C2f_CPCA module introduced in this research. The original C2f module lacked a mechanism to actively screen and enhance these key features while suppressing background noise when processing complex farmland images, causing the model to be distracted by irrelevant information and affecting segmentation accuracy. The C2f_CPCA module integrates the channel priority convolutional attention (CPCA) mechanism. This mechanism dynamically learns and generates an ‘attention map’, explicitly guiding the model on ‘where to focus’, enabling more precise extraction of discriminative features related to agricultural targets and significantly enhancing robustness in complex backgrounds. Lastly, the C3RFEM module is integrated following the SPPF module in the backbone network. Placing the C3RFEM module after the SPPF module is an advanced strategy of ‘first convergence, then enhancement’. This combination ensures that the model can extract the most powerful features from the optimal receptive field range, whether for large-scale ‘harvested areas’, medium-to-small-scale ‘agricultural machinery’ and ‘people’, or even texture-complex ‘ridge between fields’, thereby significantly improving the model’s segmentation accuracy and generalization capability for multi-scale targets.

### 2.3. Add Small Object Detection Layer

In convolutional neural networks (CNNs), feature pyramids are central to handling multi-scale object detection. The network progressively enlarges the receptive field and extracts high-level semantic features through consecutive convolutions and downsampling operations (such as convolutions with a stride of 2 or pooling). As the convolutional layers deepen, the resolution of the feature maps decreases, focusing more on global information, which is suitable for detecting large objects. In this paper, entire farmland areas include “having harvested area” and “unharvested areas,” and large agricultural machinery includes “harvesters” and tractors. Additionally, “slight lodging areas” form large, continuous irregular regions when lodging occurs in patches. When these regions occupy a prominent position in the image, their characteristics (such as shape, color, and texture) are highly distinctive. These targets typically constitute the background or primary foreground of the image, covering a wide area, and are the most typical large targets. By contrast, shallow feature maps retain more details, making them better for detecting small objects. In this article, they refer to people, etc. However, this mechanism has the following inherent drawbacks when dealing with small objects:(1)Challenges faced by the original YOLOv8n-seg in detecting small objects: The YOLOv8n model generates feature maps at three scales—80 × 80, 40 × 40, and 20 × 20—from input images of 640 × 640 resolution for detection. For extremely small objects that may only occupy a few pixels (such as distant people), after multiple downsampling steps, the effective information on low-resolution feature maps like 20 × 20 or 40 × 40 is almost completely lost, making them difficult to recognize.(2)Loss of detail information: Although shallow networks contain rich detail and edge information, which is crucial for accurately localizing small objects, they lack high-level semantic information. Deep networks have strong semantic information but suffer from low spatial resolution and coarse details. The original YOLOv8n-seg neck network partially restores some details through upsampling and fusion, but its finest fused feature map scale is only 80 × 80. For the numerous tiny targets commonly found in agricultural scenarios, the feature details at this resolution are still insufficient.(3)Scarcity of assigned anchor points: Object detection algorithms need to assign ground truth boxes to specific locations (anchor points) on the feature map for learning. Low-resolution feature map grids are sparse, and a tiny target may not find enough matching anchor points within a 20 × 20 grid, leading to missed detections (false negatives). Even when assigned, the number of grid cells used for prediction is very limited, making it difficult for the model to learn stable and robust feature representations.

Therefore, to fundamentally address the aforementioned issues, this study added a dedicated detection layer for small objects based on YOLOv8n. This improvement significantly enhances the model’s instance segmentation performance in agricultural scenarios through the following mechanisms:(1)Preserving high-resolution detailed information: The newly added P2 layer processes feature maps at a scale of 160 × 160. These feature maps are obtained by upsampling outputs from shallower network layers and have undergone fewer downsampling operations, thus retaining richer spatial details and texture information. This enables the model to “see” clearer edges and finer structures, providing a crucial information foundation for accurately segmenting and localizing small objects.(2)Providing denser prediction anchors for small objects: the 160 × 160 high resolution corresponds to a denser grid of cells. This greatly increases the chances that small objects are correctly assigned and learned, effectively reducing missed detections caused by sparse anchors. Each small object can now be predicted by more and finer grid cells working together, thereby improving detection recall and the accuracy of segmentation boundaries.(3)Optimized feature pyramid structure: The addition of the P2 layer enhances the model’s multi-scale detection capabilities. The model now features four detection scales ranging from 160 × 160 (fine), to 80 × 80 (medium), to 40 × 40 (coarse), to 20 × 20 (very coarse), forming a feature pyramid with broader coverage and smoother gradients. This design ensures that objects of any size can be detected and segmented at the most appropriate feature scale, particularly addressing the original model’s perception gap at the “extremely small” object scale.

As shown in Figure 3, in the specific implementation, in the neck network of YOLOv8, after the feature map is upsampled twice to generate the 80 × 80 scale, we added an additional upsampling operation to generate a 160 × 160 scale feature map, and we built a new detection head on this basis. This improvement enables the network to fully leverage the high-resolution advantages of shallow-layer features, specifically targeting small objects in images (such as workers in distant fields or small agricultural machinery) for efficient detection and precise segmentation, thereby comprehensively enhancing the overall performance of instance segmentation tasks in complex agricultural scenarios. The left column represents the feature layers within the backbone network, the middle column denotes the upsampling process, and the right column indicates the final generated feature map scale.

### 2.4. CPCA Attention Mechanism

In the agricultural field scene segmentation task, challenges such as target diversity, occlusion, background complexity, and scale variation are encountered. The variety of target categories can be interfered with by complex backgrounds and occlusions, making detection and segmentation tasks more difficult. The channel priority convolutional attention (CPCA) [15] mechanism effectively captures relationships at different spatial scales by dynamically adjusting attention weights in both channel and spatial dimensions, enhancing the model’s understanding of local and global structures, thereby improving detection and segmentation performance. Additionally, the CPCA mechanism is lightweight; it optimizes the attention mechanism through deep convolution and 1 × 1 convolution, adding almost no complexity to the model while reducing computational resource consumption, achieving a good balance between performance and efficiency. A schematic diagram of the CPCA attention mechanism is shown in Figure 4.

The channel priority convolutional attention (CPCA) module sequentially performs channel attention and spatial attention. For the input feature map, the channel attention module (CA) first generates a one-dimensional channel attention map, which is then element-wise multiplied with the input features. The channel attention values are broadcasted along the spatial dimensions to obtain refined features with channel attention. Next, the spatial attention module (SA) generates a three-dimensional spatial attention map, and the final output features are obtained by element-wise multiplication. The overall process is as follows:(1)$$\begin{array}{c} F_{c}=CA\left(F\right)⊗F \end{array}$$(2)$$F_{b}=SA\left(F_{c}\right)⊗F_{c}$$

Among them, ⊗ represents element-wise multiplication, and F represents the input feature map (with shape C × H × W).

$CA\left(F\right)\;denotes\;the\;channel\;attention\;weight\;map\;(with\;shape\;C\times 1\times 1),\;F_{c}$ is the feature map enhanced by channel attention $SA\left(F_{c}\right)$, denoting the spatial attention weight map (with shape 1 × H × W), $and\;F_{b}$ is the final feature enhanced by both channel and spatial attention.

The workflow of Equation (1) is as follows: First, the importance weights of each channel are computed (via the CA module). The weight map is then broadcasted to the shape C × H × W and element-wise multiplied with the original feature map. This process enhances important channels while suppressing irrelevant ones.

The workflow of Equation (2) is as follows: First, the spatial importance is computed based on the input feature map (via the SA module) to generate a spatial weight map (in the form of a heatmap). This weight map is then element-wise multiplied with the feature map, thereby enhancing key regions and attenuating background noise.

The channel attention map is generated by the channel attention module. By performing average pooling and max pooling operations on the feature map, spatial context descriptors are obtained. These descriptors are input into a shared multi-layer perceptron (MLP), and the channel attention map is obtained by element-wise summation of the MLP’s output. The calculation formula is expressed as follows:(3)$$\begin{array}{c} CA\left(F\right)\;=\;\sigma (\mathrm{MLP}(\mathrm{AvgPool}\left(F\right))+MLP(\mathrm{MaxPool}\left(F\right))) \end{array}$$

Spatial attention maps capture spatial relationships through deep convolution and enhance the capability of convolution operations with multi-scale structures. Finally, channel mixing is performed through 1 × 1 convolution at the end of the module. The calculation formula is expressed as follows:(4)$$\begin{array}{c} SA\left(F\right)={Conv}_{1\times 1}(\sum_{i=0}^{3}{\mathrm{Branch}}_{i}(\mathrm{DwConv}(F))) \end{array}$$

Among them, DwConv represents deep convolution, and Branch_i_ represents the i-th branch (Branch_0_ is an identity connection).

### 2.5. Neck Network Improvement

The neck structure of YOLOv8 adopts a PAN-FPN design, which achieves efficient fusion of multi-scale features by combining bottom–up and top–down feature propagation, significantly enhancing the model’s detection accuracy and adaptability. Among them, the PAN structure [16] enhances the expressive capability of the feature maps, while the improved FPN structure [17] increases feature extraction efficiency by replacing the traditional C3 module with the C2f module.

The C2f module enhances the performance of object detection and multi-scale feature extraction capabilities by merging low-level and high-level feature maps, while also reducing information loss and strengthening the model’s nonlinear representation ability and scalability. However, this design’s multi-path processing strategy also introduces a certain level of complexity. Specifically, the C2f module divides the input feature map into two branches, with one branch directly passing to the output and the other branch being processed through a bottleneck module. Ultimately, these feature maps (including the directly passed feature maps and the processed feature maps) are concatenated together and further processed by subsequent layers. The output feature map consists of two directly processed feature maps and n feature maps processed by the bottleneck module, where n represents the number of bottleneck modules.

By contrast, the C2f_CPCA module proposed in this paper has some optimizations in its design. Unlike the C2f module, the C2f_CPCA module no longer directly retains the feature map from the first branch but instead processes it first through a channel attention mechanism (CPCA channel attention). The feature map processed by the attention mechanism is then concatenated with other feature maps processed by the bottleneck module to form the final output feature map. This output consists of one feature map processed by the attention mechanism and n feature maps processed by the bottleneck module.

The C2f_CPCA module was proposed to optimize the feature expression capability of the C2f module while reducing the number of model parameters and computational load. By introducing a channel attention mechanism (CPCA) into the traditional C2f structure, the C2f_CPCA module improves the accuracy of feature selection, allowing the model to focus more on key features. Furthermore, this design retains the advantages of multi-scale feature fusion and simplifies computation by eliminating directly passed branches, enabling the module to enhance expression capability while maintaining low resource consumption.

This study aims to improve the neck network structure by replacing the C2f module in the neck network with the C2f_CPCA module in order to optimize the model’s accuracy in agricultural scene segmentation while reducing the model’s parameter count and computational load. As shown in Figure 5, this paper proposes two ways to apply the C2f_CPCA module to the neck network, which includes small object detection layers. Figure 5a displays the original neck network structure; in Figure 5b, the C2f_CPCA module replaces four C2f modules that connect to the segmentation head; and in Figure 5c, the C2f_CPCA module replaces all C2f modules in the neck network. Experimental results indicated that the neck structure shown in Figure 5c not only further enhanced the model’s segmentation performance but also reduced computational overhead.

### 2.6. Backbone Network Improvement

In order to further enhance the feature extraction capability of the model backbone network and improve the segmentation accuracy of multi-scale targets in complex agricultural scenarios, the RFEM [18] (Receptive field enhancement) module was introduced into the backbone network of YOLOv8 in this study.

Figure 6 shows the structure of C2f_CPCA and C2f modules.

As shown in Figure 7a, the basic structure of the C3 module is shown. The C3 module is a key component introduced in the YOLOv5 core network. Its name is derived from the combination of the CSP (cross stage partial) architecture and three convolutional layers (conv). It serves as an efficient, foundational feature extraction unit that enhances gradient flow and information flow.

As shown in Figure 7b, the RFEM module consists of two parts: one is a multi-branch based on extended convolution, and the other is a weighted layer (as shown in Figure 6). In the multi-branch part, different expanded convolutions with expansion rates of 1, 2, and 3 are used, and all convolutions use a fixed 3 × 3 convolution kernel size. By introducing residual connections, gradients can be prevented from exploding and disappearing during training, thus improving the detection accuracy of the model and reducing information loss during feature extraction [19,20].

Since the C2f module already has a high parameter count and computational complexity, the RFEM module was integrated into the C3 module. As the core of the YOLOv5 backbone network, the C3 module has stronger scalability and stability and can better support the enhancement of multi-scale feature extraction capability brought by the RFE module.

As shown in Figure 7c, based on the bottleneck structure in the C3 module, the C3RFEM module is formed after replacing the original bottleneck module with the RFEM module. The SPPF module in the YOLOv8 backbone network integrates the feature map information through multi-scale pooling. In this study, the C3RFEM module is placed after the SPPF module to further deepen the depth of the backbone network, so as to extract more abundant features in different receptive fields. Experiments showed that this design not only improved the detection accuracy and robustness of the model for multi-scale targets in complex agricultural scenarios but also achieved better feature extraction performance without significantly increasing the computational burden.

## 3. Experiments and Analysis

### 3.1. Data Set

The dataset of this study was composed of agricultural scene images collected manually and annotated using LabelImg software (Python 3.10). The dataset was collected in May 2024 at a test field in Xiaoshan, Hangzhou, Zhejiang Province, China, using an Intel RealSense D415 depth camera with a depth range of 0.3 m to 10 m. The color image resolution was 1920×1080, and the depth image resolution was 1280 × 720. The camera was mounted on a Kubota harvester, and data were collected during three time periods: early morning, midday under strong sunlight, and dusk. The original dataset contained 593 images. In order to enhance the diversity of the data and better adapt to the actual application scenario, each original image was subjected to data enhancement processing, including linear contrast enhancement, Gaussian noise addition, and horizontal flipping, and the dataset was finally extended to 2372 images. The dataset was divided into a training set, verification set, and test set in a ratio of 7:2:1. The dataset included the following categories: ridge between fields, having harvested areas, obstacle, slight lodging areas area, harvester, people, unharvested area, and farm. The specific sample sizes for each instance are shown in Table 1. The data augmentation parameters were optimized as follows: The adjustment coefficient for linear contrast enhancement was set to 1.5 to simulate varying lighting conditions. The standard deviation of Gaussian noise was set to 0.1 to mimic the random noise from image acquisition devices. Horizontal flipping was applied to 50% of the samples to enhance the model’s adaptability to changes in perspective. These processes not only expanded the scale of the dataset but also significantly improved its representativeness in complex environments, providing a solid foundation for subsequent model training and evaluation.

From the perspective of the number of categories in the dataset, there was a category imbalance issue (e.g., the proportion of people was less than 5%). To address the category imbalance in the dataset, this study adopted a weighted loss function strategy during the training phase. Specifically, when calculating the segmentation loss (such as dice loss or cross-entropy loss), a weight (w_c_) was assigned to each category (c). This weight was inversely proportional to the frequency of the category, typically calculated based on the number of valid samples or using the sqrt function. For minority categories, such as “people” and “obstacle,” the weight was significantly higher than that of majority categories, such as “unharvested areas.” This forced the model to focus more on minority category samples that were difficult to classify correctly during training. The calculation formula is as follows:(5)$$Loss=-\Sigma \left(w_{c}*y_{true}*log\left(y_{pred}\right)\right)$$
where w_c_ = 1/log(frequency_c_ + α), where the parameter α is a smoothing hyperparameter that prevents numerical instability and controls the magnitude of weight updates, and frequency_c_ is the frequency of category c in the training set; y_true_ is the true label, which is an encoded vector; and y_pred_ represents the predicted probability, and the sum of the probabilities of all categories is 1.

To gain a deeper understanding of how these enhancement methods affected the characteristics of farmland images, this study conducted systematic quantitative analysis. Figure 8 demonstrates the visual differences between the original farmland scene and its enhanced variants. The original image (a) shows a typical farmland landscape; contrast enhancement (b) significantly improves the contrast between bright and dark areas, making crop-soil boundaries clearer; Gaussian noise addition (c) introduces granular interference to simulate sensor noise or image degradation under low-light conditions; and horizontal flipping (d) retains all visual features while only altering the spatial orientation.

Figure 9 demonstrates how different enhancement methods alter image statistical characteristics through pixel value distribution curves. This figure shows a smoothed curve representing the statistical histogram of image pixel brightness values (0–255). The horizontal axis represents pixel brightness (from 0 pure black to 255 pure white), and the vertical axis represents the frequency of pixels with specific brightness values in the image (normalized). After the model has been trained on a more diverse brightness distribution, its robustness to extreme lighting changes in real environments is significantly enhanced. Gaussian noise simulates the noise introduced by harsh transmission environments, enabling the model to learn segmentation under mild noise interference and reducing the risk of performance degradation caused by imperfect image quality. Figure 10’s gradient amplitude distribution quantifies the evolution patterns of edge features. It first uses operators such as Sobel to calculate the gradient magnitude of each pixel (measuring the ‘strength’ of that point belonging to an object edge) and then plots the frequency distribution of these magnitudes. The horizontal axis represents the gradient magnitude (ranging from 0 to high variation), while the vertical axis shows the frequency of occurrence for each magnitude. One of the core challenges in instance segmentation is generating precise object mask boundaries. After contrast enhancement, edge signals become stronger, providing the model with higher-quality learning signals while also amplifying the feature signals of small target. While Figure 11’s PCA-based feature space analysis reveals deep-level color variations, it uses principal component analysis (PCA), a dimension reduction technique, to compress the high-dimensional depth features of each image (e.g., features extracted by a pre-trained network) into the two most representative dimensions (PC1 and PC2), with each point representing an image. The PCA graph intuitively proves that the enhancement strategy effectively expands the feature coverage of the training set and reduces the risk of overfitting. This multidimensional visualization approach not only validates the effectiveness of the enhancement strategies but also provides theoretical foundations for improving model performance in real-world challenges such as illumination changes, sensor noise, and viewing angle differences.Blue lines and blue dots represent the original image; Orange lines and orange dots represent the image with enhanced contrast; Green lines and green dots represent the image with added Gaussian noise; Red lines and red dots represent the horizontally flipped image.

### 3.2. Evaluation Index

In the experiments in this paper, the evaluation index of the improved YOLOv8 instance segmentation model included two tasks: object detection and segmentation. However, this paper focuses on the accuracy of instance segmentation, so only the evaluation index of instance segmentation was used in the subsequent experiments. In the instance segmentation task, the indicators to evaluate model performance included accuracy (P_M_), recall rate (R_M_), and average accuracy (mAP_M_). The subscript M represents Mask, which is used to identify the instance segmentation task. In terms of computation, parameters and GFLOPs were used to measure the parameter size and computational complexity before and after model improvement. Some of the evaluation index formulas are listed below:(6)$$\begin{array}{c} P_{M}=\frac{TP}{TP+FP} \end{array}$$(7)$$\begin{array}{c} R_{M}=\frac{TP}{TP+FN} \end{array}$$(8)$$\begin{array}{c} AP_{M}=\int_{0}^{1}P_{M}\left(R_{M}\right)\;d{(R}_{M}) \end{array}$$(9)$$\begin{array}{c} mAP_{M}=\frac{1}{N}\sum_{i=1}^{N}{\left({AP}_{M}\right)}_{i} \end{array}$$
where accuracy (P_M_) refers to the proportion of positive samples predicted by the model that are actually positive samples, and recall rate (R_M_) refers to the proportion of correctly predicted positive samples in actual positive samples. AP_M_ is the area under the precision–recall curve used to evaluate the segmentation performance of the model for a single class, showing the average accuracy of the model under different recall rates. Average accuracy (mAP_M_) is used to calculate the average AP_M_ value of all classes to reflect the overall accuracy of the model on the instance segmentation task, and the arithmetic average of all classes of AP_M_ values is used to comprehensively evaluate the segmentation performance of the model on the entire dataset. For consideration of the intersection ratio (IoU), mAP0.5_M_ calculates the average accuracy at an IoU threshold of 0.5, representing the model performance when the segmentation result overlaps with the true annotation by more than 50%. mAP0.5:0.95_M_ combines the average accuracy of different IoU thresholds from 0.5 to 0.95, providing a comprehensive performance evaluation of the model under different overlap requirements.

### 3.3. Experimental Environment

The experiments in this paper were carried out on the Windows 10 operating system, using the PyTorch (version: 2.3.0) deep learning framework. The software environment was CUDA 10.2. In terms of hardware configuration, the i7-13650HX and NVIDIA GeForce RTX 4060 Laptop GPU were used. In the training process, YOLOv8n-seg was used as the benchmark model, with the batch size set to 8 and thread number set to 8. In order to prevent overfitting, the number of training iterations was set to 300 times; the number of early stops (patience) was set to 100; loss weights: class = 0.5, box = 7.5, mask = 2.5; input resolution: 640 × 640; initial learning rate: 1 × 10^−3^ (cosine annealing decay); optimizer: AdamW (β1 = 0.9, β2 = 0.999); weight decay: 5 × 10^−4^; gradient clipping norm: 10.0; and the other parameters were set by default by YOLOv8.

### 3.4. Experimental Result

#### 3.4.1. Ablation Experiment

In this paper, three optimization schemes are proposed for the YOLOv8n-seg model: firstly, detection layer P2 is added specifically for small targets; secondly, all C2f modules are replaced with the C2f_CPCA module proposed in this paper in the neck network; and finally, C3RFEM modules are added to the SPPF module in the backbone network. In order to verify the effectiveness of the improved scheme on the YOLOv8n-seg model, an ablation experiment was conducted using the evaluation index of the validation set. Under the same experimental conditions, YOLOv8n-seg was adopted as the benchmark model, and its training results were taken as the benchmark. By gradually applying these improvements to the YOLOv8n-seg model, the ablation study explored the influence of different improvement combinations on the model segmentation accuracy. The evaluation indexes of the experimental process included algorithm parameters, floating-point operations per billion (GFLOPs), accuracy (P_M_), recall rate (R_M_), mAP0.5_M_, and mAP0.5:0.95_M_. In the ablation experiment, a set of comparison experiments was also conduced, in which the four C2f modules connected to the segmentation head were replaced with the C2f_CPCA modules, which was identified as YOLOv8n-seg + P2 + C2f_CPCA_1.

According to Table 2 of the experimental results, after the introduction of small target detection layer P2, the computational load of the model was increased to 26.0 GFLOPs, the number of parameters was slightly reduced, the accuracy was increased to 0.922, the recall rate was increased to 0.911, and mAP@0.5 and mAP@0.5:0.95 were 0.954 and 0.698, respectively. When the C2f_CPCA module was introduced into the network, the number of parameters was further reduced to 3,133,056, and the computational load was slightly reduced to 25.9 GFLOPs. Although the accuracy decreased slightly to 0.914, the recall rate increased significantly to 0.927, and mAP@0.5 and mAP@0.5:0.95 increased to 0.958 and 0.705, respectively. This showed that the C2f_CPCA module was lighter in parameters and computational load and the feature extraction and recognition abilities of the model were further improved.

The comparison experiment YOLOv8n-seg + P2 + C2f_CPCA_1 showed that although the model was close to YOLOv8n-seg + P2 + C2f_CPCA in terms of parameter number and calculation amount, the accuracy and mAP index decreased. This showed that it was more reasonable to replace all of the C2f_CPCA modules in the neck network rather than only part of the C2f modules.

In the final improvement plan, three improvement methods were applied at the same time. mAP@0.5 and mAP@0.5:0.95 were increased to 0.959 and 0.711, respectively, and the recall rate was increased to 0.932. Although the number of parameters increased to 3,347,200, it was still within reasonable range, and the computation volume remained at 26.0 GFLOPs. Compared with the benchmark model YOLOv8n-seg, the accuracy (P_M_), recall rate (R_M_), mAP@0.5, and mAP@0.5:0.95 indexes of the final improved model had improved by 1.4%, 4.0%, 3.0%, and 3.5%, respectively.

The performance gain of the CPCA + C3RFEM model over the original YOLOv8n-seg + P2 + C2f_CPCA configuration (mAP@0.5: from 0.705 to 0.711) may appear modest, but this improvement carried substantial practical and theoretical significance:(1)Fundamental enhancement in multi-scale feature extraction: The C3RFEM module’s multi-scale receptive fields significantly boost the model’s modeling capacity for complex structures and contextual information—capabilities that neither the attention mechanism (C2f_CPCA) nor the shallow P2 layer alone could fully achieve.(2)Complementary synergy with attention mechanisms: While C3RFEM generates richer feature maps, C2f_CPCA excels at extracting critical information from these features with precision. This combination achieves a balance between “expanding resources” and “conserving computation,” leading to further performance gains.

Although this gain resulted in a 6.6% increase in parameters, it not only improved overall accuracy but, more importantly, enhanced segmentation performance for challenging agricultural categories like field ridges and human figures—a crucial improvement for real-world applications. With significantly lower parameter requirements and computational load compared to other models achieving equivalent performance, the model perfectly meets deployment demands on resource-constrained agricultural embedded devices.

In order to visually show the improvement effect, this study uses a radar chart, as shown in Figure 12. In this diagram, ↓ indicates that the model has become lighter and more efficient, representing a positive improvement.

Table 3 shows the results of the comparison between YOLOv8n-seg and the algorithm in this paper on the detection performance of various categories on the test set. It can be obviously seen that the algorithm in this paper significantly improved the average accuracy of all categories. Especially in the ridge between fields category, the recall rate (R_M_), average accuracy (mAP@0.5_M_), and average accuracy (mAP@0.5:0.95_M_) were increased by 19.8%, 13.8%, and 9.2%, respectively. In the people category, the corresponding increases were 9.5%, 6.6%, and 4.7%. Although the accuracy of the proposed algorithm was similar to that of the original algorithm in some categories, such as having harvested areas and harvesters, the significant improvement in recall rate of the improved model makes it more reliable in practical agricultural scenarios. The promotion of high recall rates is particularly important in practical agricultural applications because it directly affects the overall identification of targets and the accuracy of operations. For example, accurate identification of the ridge is essential to distinguish the field and carry out the operation, and a high recall rate can effectively avoid the operation error caused by missed detection. Similarly, for having harvested areas, accurate identification can help to optimize subsequent operational planning and resource allocation, thereby improving the efficiency and precision of agricultural production. In addition, when detecting obstacles, high recall rates can more fully identify potential risks in operations and reduce the probability of accidents. In terms of people detection, a high recall rate ensures that all personnel on the job are accurately identified, which is critical for safety management and monitoring of job progress.

In order to verify the detection effect of the improved algorithm in practical application, this study selected a number of images in the test set for testing and compared the results with the original algorithm. The segmentation results are shown in Figure 13. The three pictures on the left show the segmentation and detection results of YOLOv8n-seg, the three pictures in the middle are the segmentation results of the proposed algorithm, and the pictures on the right are the segmentation mask of the original label of the dataset. The results of the comparative analysis show that the YOLOv8n-seg algorithm had shortcomings in segmentation accuracy, it showed certain deviations when compared with the actual label edge, and some incomplete phenomena appeared. However, the segmentation accuracy of the proposed algorithm was higher, and it could fit the real label more closely. In addition, the original algorithm still had some misrecognition problems, such as misidentifying the distant background as the category “ridge between fields,” while the proposed algorithm could effectively eliminate background interference. In terms of dealing with small targets, the comparison graph in the figure further showed that YOLOv8n-seg could not identify distant persons, while the algorithm in this paper could accurately detect and segment distant persons. Moreover, in terms of the integrity of the segmentation results, the effect of the YOLOv8n-seg algorithm was not as accurate as that of this algorithm. These results showed that the proposed algorithm has better performance in processing the image segmentation task in actual agricultural scenes.

#### 3.4.2. Algorithm Comparison Experiment

In order to further reflect the light weight and high efficiency of the proposed algorithm, it was compared with other segmentation algorithms of the YOLOv5 series, YOLOv7, YOLOv8n, YOLOv8s, and YOLOv9t, and the experimental results are shown in Table 4. The mAP0.5M index of our method reached 0.959, which was significantly better in average accuracy compared with lightweight models such as YOLOv5n, YOLOv5s, YOLOv8n, YOLOv9t, YOLOv10n, YOLOv10s, Mask R-CNN, and Mask2Former and even better than some lightweight models with relatively large parameters and computational load, such as YOLOv5m and YOLOv7. This indicated that in the instance segmentation task of agricultural scenarios, a model with a large number of parameters may have redundancy, leading to overfitting of training data. Models with smaller parameter numbers are better able to capture major trends in the data and reduce overfitting of outliers or noise, thus showing more stable and reliable performance on new data.

In addition, although compared with YOLOv8s the proposed algorithm was not superior in the mAP0.5:0.95_M_ index, it had a lower parameter number (3,347,200) and computational complexity (26.0 GFLOPs), which is more convenient for practical application and deployment. It showed that the proposed algorithm achieved a good balance between computation cost and detection segmentation performance. Compared to the latest YOLOv10, YOLOv10n demonstrated superior performance among lightweight models, yet our model still slightly outperformed it in mAP@0.5 (0.959 vs. 0.935), indicating the effectiveness of our improvement strategy. While YOLOv10s achieved slightly higher accuracy than ours, it required significantly more parameters and computational resources, making our model more efficient in balancing accuracy and performance. When compared with Mask2Former, which achieved the highest accuracy but consumed over ten times more computations than ours, our model maintained high precision while ensuring real-time processing, making it more suitable for embedded deployment in agricultural applications. In order to show the contrast effect more intuitively, this study uses a bubble chart of multi-model performance comparison, as shown in Figure 14.

The proposed improved YOLOv8n-seg model achieved an excellent balance between computational complexity and detection performance: with a parameter size of only 3.35 million and computational cost of 26.0 GFLOPs, it delivered real-time inference speeds of approximately 9.2 milliseconds (108 FPS) on the RTX 4060 GPU. Notably, this model demonstrated outstanding accuracy, achieving mAP@0.5 of 0.959—surpassing lightweight models like YOLOv5s and YOLOv8n while approaching the performance of more computationally intensive variants such as YOLOv8s and Mask R-CNN. This design philosophy of “high efficiency, lightweight computation, and uncompromised accuracy” makes it particularly suitable for deployment in agricultural embedded devices with limited computing resources.

#### 3.4.3. Model Generalization Analysis and Discussion

Although the improved model performed excellently on our own test set, we designed additional experiments to verify its generalization ability in order to ensure its reliability in practical applications. We selected 187 images from the public agricultural image dataset AgroSeg, which contained similar categories (such as farmland, crops, and people) but were collected from different regions and scenes, forming an external test set. On this dataset, we directly tested our trained final model (YOLOv8n-seg + P2 + C2f_CPCA + C3RFEM) without any fine-tuning and compared it with the original YOLOv8n-seg model.

As shown in Table 5, the proposed model achieved 4.2% and 4.1% improvements in mAP@0.5 and mAP@0.5:0.95 metrics on an external test set from unknown domains compared to the original model. This demonstrated that the proposed enhancements (small object detection layer, CPCA attention mechanism, and C3RFEM module) not only improved the model’s performance on existing datasets but also significantly boosted its ability to extract features and adapt to new scenarios and data—essentially enhancing its generalization capabilities. The attention mechanism and multi-scale feature enhancement enabled the model to better focus on the essential features of objects rather than overfitting specific backgrounds in the training set.

It should be noted that both models exhibited significantly lower absolute performance on the external test set compared to their own training sets (mAP@0.5 decreased from 0.959 to 0.683). This decline was expected, primarily due to three factors: (1) Domain Gap: Fundamental differences between external test sets and the training set in crop types, cultivation patterns, soil coloration, and climatic conditions. (2) Annotation Variations: Potential subtle discrepancies in labeling standards between public datasets and this study. (3) Sample Size Limitations: Despite data augmentation efforts, the limited sample size of approximately 1600 training images remained insufficient for adequately covering diverse agricultural scenarios, which constitutes the primary bottleneck affecting the model’s generalization capability.

Experimental results demonstrated that the proposed model exhibited superior generalization performance compared to the baseline model, though there remains room for further improvement in absolute performance metrics. This finding highlights two critical research priorities: first, collecting larger-scale and more diverse agricultural scenario data to build a comprehensive benchmark dataset; and second, developing domain adaptation techniques to enhance model transferability across different agricultural environments, thereby meeting the demands of smart agriculture applications.

## 4. Discussion

### 4.1. Limitations and Challenge Scenario Analysis

While the proposed improved YOLOv8n-seg model demonstrated significant performance enhancements across overall metrics and multiple categories—particularly excelling in small targets (e.g., people) and challenging categories (e.g., ridge between fields)—it still faces practical deployment limitations and performance bottlenecks in specific scenarios, as detailed in Table 5. Analyzing these issues helps to clarify the model’s applicability boundaries and guides future optimization efforts.

#### 4.1.1. Robustness Under Extreme Light Conditions

Agricultural scenes exhibit extreme lighting variations (such as intense midday sunlight, low twilight illumination, and backlighting). While data augmentation techniques (e.g., contrast adjustment) provide some assistance, the model’s detection and segmentation accuracy may be significantly degraded under extreme lighting conditions (e.g., severely overexposed or underexposed areas). The CPCA mechanism’s reliance on feature maps could amplify noise or obscure critical details under these conditions. Additionally, the model may exhibit false detections (mistaking light and shadow for targets), missed detections (where targets blend into the background), or blurred segmentation boundaries. For example, “field ridges” might be difficult to distinguish from soil under strong shadows, while “unharvested areas” could be misidentified as “having harvested areas” in backlit scenarios. Improvement directions: Explore more robust light-invariant feature extraction methods, incorporate broader extreme lighting samples into training datasets, and even consider integrating information from other sensors (such as near-infrared).

#### 4.1.2. Intense Occlusion and High Overlap of Targets

In agricultural fields, crops, weeds, farm machinery, and personnel often overlap or partially block each other (e.g., weeds in dense crops, people behind farm equipment, or crops caught by harvesters). Current models primarily rely on single-frame image data. In severe occlusion scenarios, the model may fail to detect fully occluded targets (low recall) or only segment partial visible areas (incomplete segmentation masks). For highly overlapping similar objects (such as dense weed clusters), instance segmentation may struggle to accurately distinguish individual elements (mask adhesion), leading to imprecise instance counting. Improvement directions: Introduce temporal information (using video sequences) for tracking and occlusion inference, explore more powerful contextual modeling and graph neural networks (GNN) to handle object relationships, and refine loss functions to encourage clearer instance boundary segmentation.

#### 4.1.3. Long-Range Very Small Target Detection

While the newly added P2 layer significantly enhances small object detection capabilities, distant targets in images (such as personnel at the far edge of fields or small obstacles) may occupy minimal pixel areas (e.g., less than 10 × 10 pixels), resulting in extremely weak feature information. The recall rate for these small targets could be substantially lower than that of nearby or medium-distance objects, leading to potential missed detections. Even when detected, the segmentation mask accuracy (IoU) tends to remain low. Improvement directions: Consider higher-resolution inputs (increasing computational costs), design more sophisticated feature fusion strategies to fully leverage shallow high-resolution features, explore super-resolution techniques or specialized small target attention mechanisms, and integrate target tracking to predict potential small target positions.

#### 4.1.4. Limitations of Model Generalization Ability

While the model demonstrated excellent performance when trained and evaluated on specific datasets (the terraced rice scenario in this study), with improved generalization ability across external test sets, we must acknowledge significant differences in scenario characteristics across crop types (e.g., wheat and corn), cultivation patterns (e.g., open-field farming and greenhouse systems), topography (e.g., plains and hills), and seasonal factors (e.g., winter fallow periods). Direct application of the model to other agricultural scenarios may result in substantial performance degradation, as evidenced by the performance analysis of the Challenge Scenario 5 shown in Table 6. Improvement directions: Construct larger and more diverse agricultural scenario datasets, explore domain adaptation or meta-learning techniques to enhance cross-scenario generalization capabilities, and develop modular designs that allow fine-tuning specific components for different scenarios.

## 5. Conclusions

This paper presents an instance segmentation algorithm for agricultural scenarios based on the improved YOLOv8n-seg model, designed to enhance target detection and segmentation performance in complex agricultural scenes. Ablation experiments demonstrated that the enhanced model achieved 1.4% and 4.0% mAP0.5_M_ improvements in accuracy and recall rate compared to the baseline YOLOv8n-seg model, with mAP0.5:0.95_M_ showing 3.0% and 3.5% increases, respectively. Notably, the improved model demonstrated superior average precision across multiple categories, particularly in field ridges and human targets, where significant accuracy gains substantially enhanced comprehensive recognition and segmentation capabilities. Test results indicated that the refined algorithm can more accurately segment complex objects in real-world agricultural scenes, reduce false detections, and effectively handle small targets. In comparative experiments with mainstream instance segmentation algorithms, our method outperformed other models by achieving an mAP0.5_M_ of 0.959—demonstrating clear advantages over lightweight counterparts including the YOLOv5 series, YOLOv7, YOLOv8n, YOLOv9t, YOLOv10n, YOLOv10s, Mask R-CNN, and Mask2Former. Although slightly lower than YOLOv8s in mAP0.5:0.95_M_ metrics, our algorithm achieved better parameter efficiency (3,347,200 parameters) and computational complexity (26.0 GFLOPs), striking a favorable balance between computational cost and detection performance. Compared to YOLOv10, our model still demonstrated a slight edge in mAP@0.5 (0.959 vs. 0.935). While YOLOv10s achieved slightly higher accuracy, our model outperformed it in efficiency–precision balance due to its larger parameter size and computational requirements. When compared with Mask2Former, our model maintained high precision while ensuring real-time performance, making it more suitable for embedded deployment. The model performed exceptionally well on the specified datasets and common farmland scenarios, particularly showing significant improvements in recall rates for small targets and key categories like field ridges and personnel. However, challenges remain under extreme lighting conditions, severe occlusion, long-range tiny targets, inter-class similarity confusion, and cross-scenario generalization. In overexposed/backlit scenes, mAP decreased by over 7%, necessitating the design of illumination-invariant feature extraction modules. Insufficient geometric deformation modeling requires introducing deformable convolutional architectures and spatiotemporal contextual modeling for crop lodging and agricultural machinery occlusion scenarios. Future work will focus on addressing these limitations through enhancing light robustness, utilizing temporal information for occlusion handling, optimizing tiny target detection, exploring high/low-resolution dual-channel feature fusion architectures, improving feature discriminative power, and developing domain-adaptive methods to advance agricultural instance segmentation technology for more reliable applications in complex real-world environments.

## Figures and Tables

**Figure 1 sensors-25-05506-f001:**
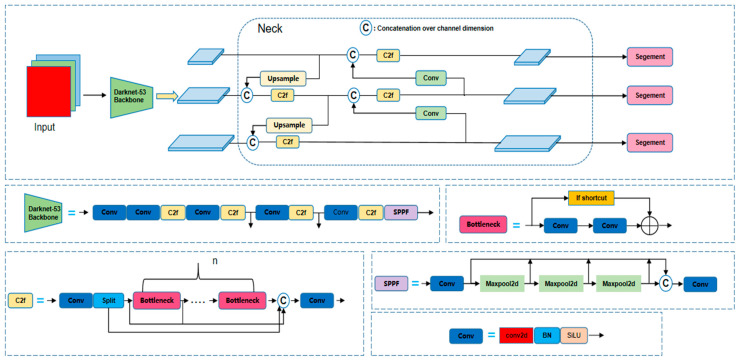
YOLOv8-seg model structure.

**Figure 2 sensors-25-05506-f002:**
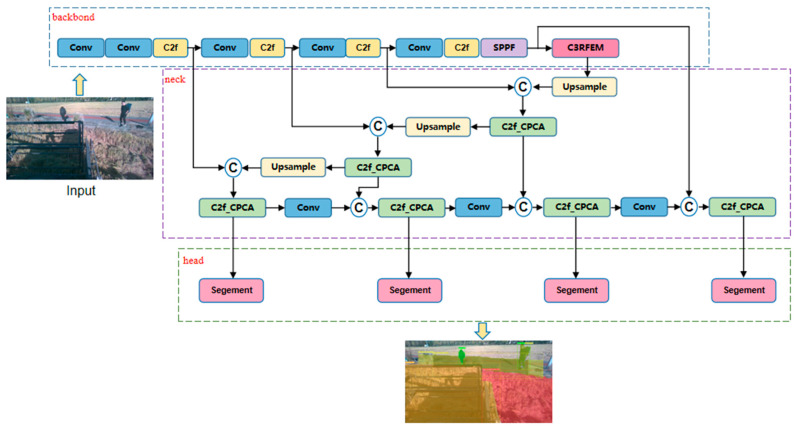
Improved YOLOv8-seg model structure.

**Figure 3 sensors-25-05506-f003:**
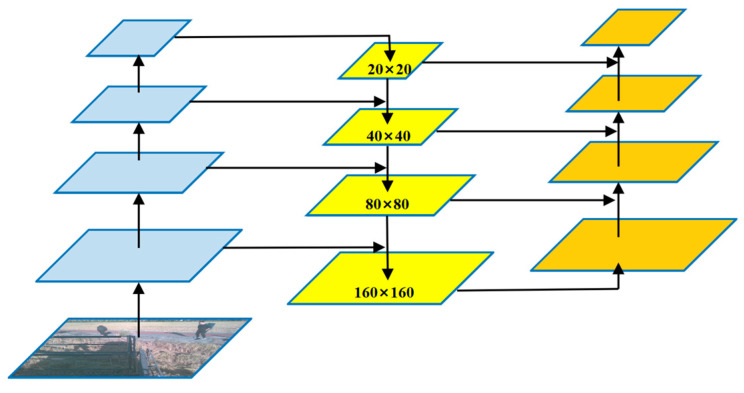
Addition of small object detection layer.

**Figure 4 sensors-25-05506-f004:**
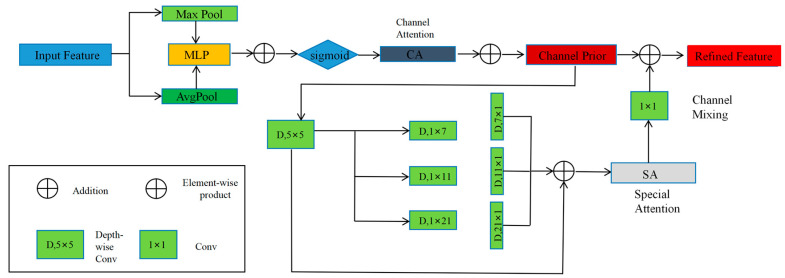
CPCA attention mechanism diagram.

**Figure 5 sensors-25-05506-f005:**
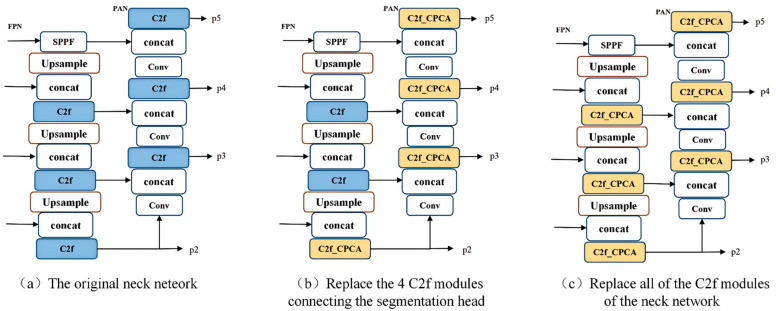
C2f_CPCA module applied to the neck network.

**Figure 6 sensors-25-05506-f006:**
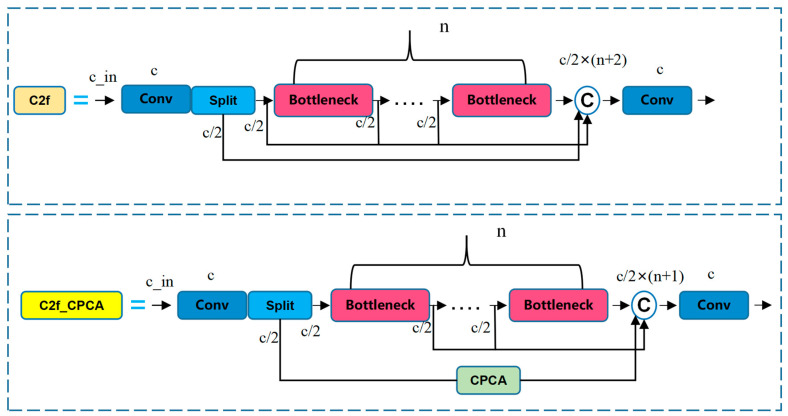
C2f_CPCA and C2f module structure diagram.

**Figure 7 sensors-25-05506-f007:**
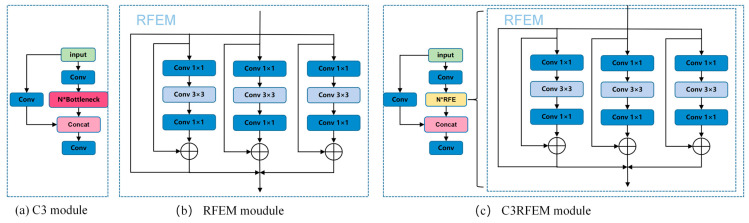
C3, RFEM, and C3RFEM module structure diagram.

**Figure 8 sensors-25-05506-f008:**
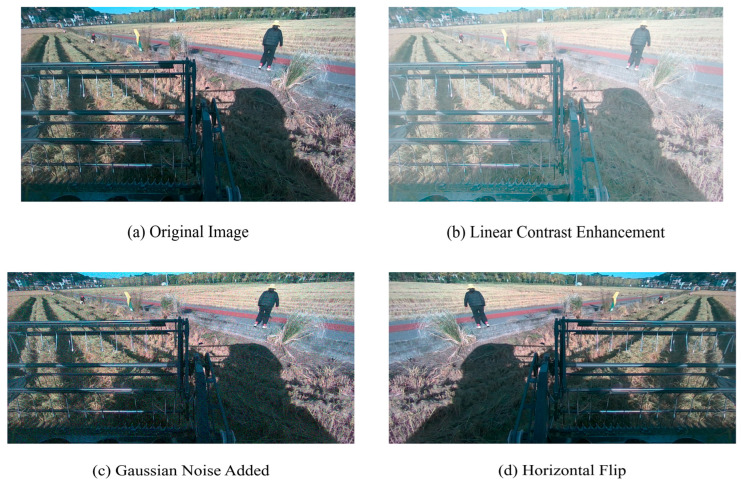
Visual difference comparison of dataset images.

**Figure 9 sensors-25-05506-f009:**
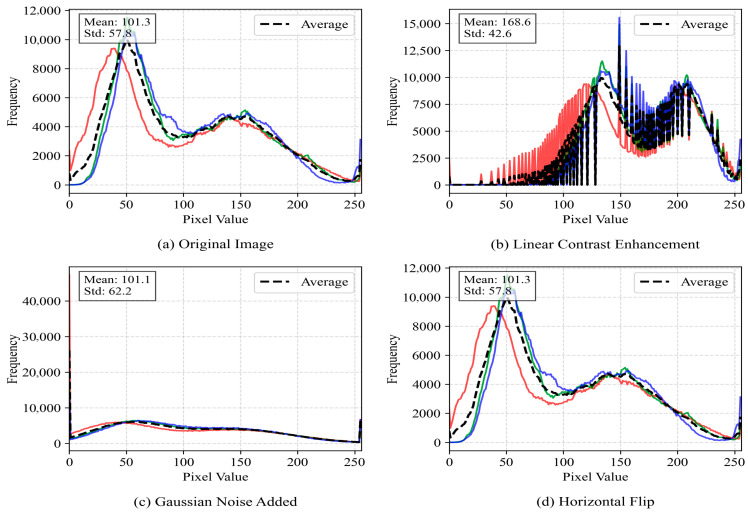
Comparison of differences in pixel value distribution curves.

**Figure 10 sensors-25-05506-f010:**
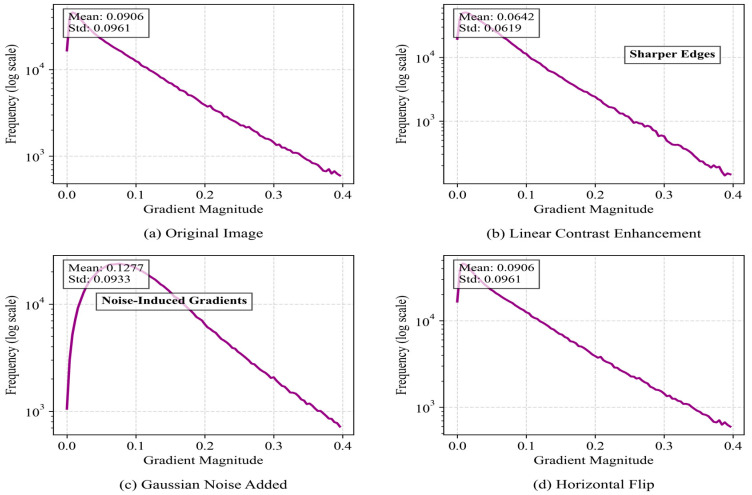
Comparison of edge characteristic curves.

**Figure 11 sensors-25-05506-f011:**
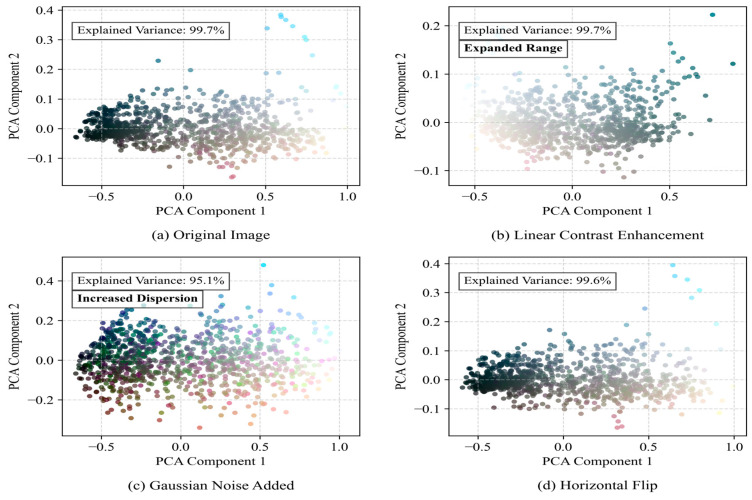
Comparison of PCA-based feature space distribution differences.

**Figure 12 sensors-25-05506-f012:**
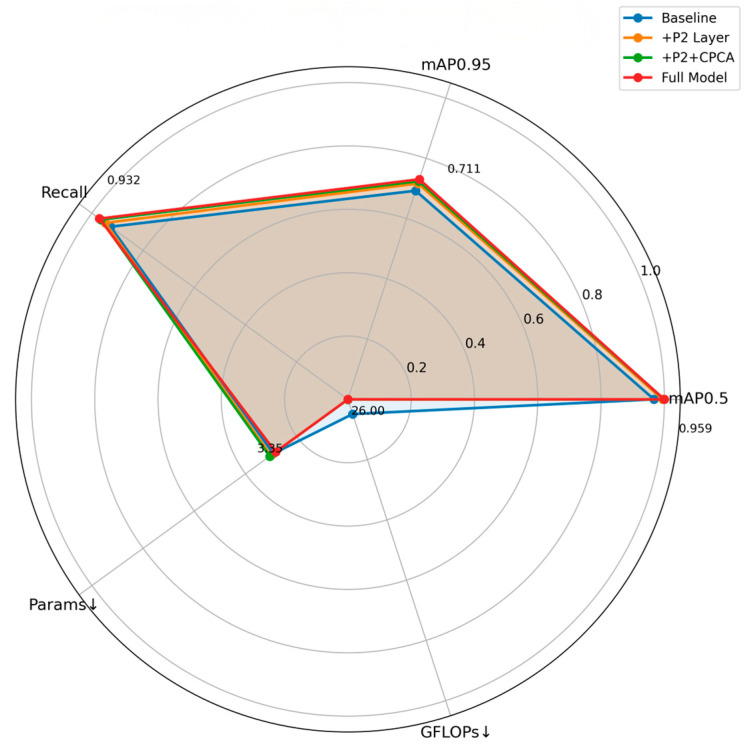
Radar map of ablation experiment.

**Figure 13 sensors-25-05506-f013:**
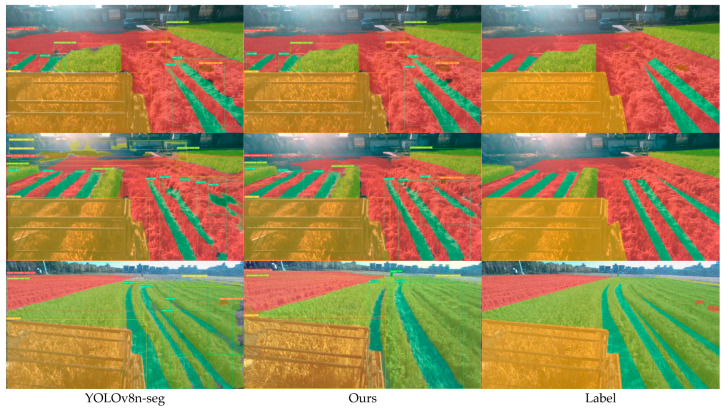
Comparison of detection results between the proposed algorithm and YOLOv8n-seg.

**Figure 14 sensors-25-05506-f014:**
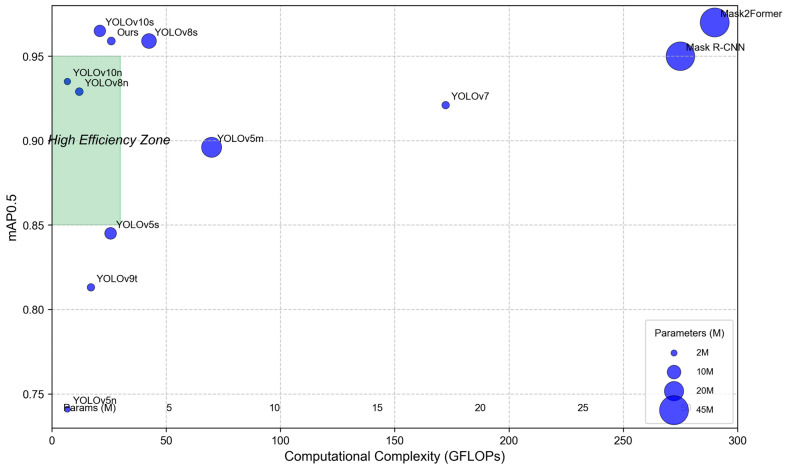
Multi-model performance comparison bubble chart.

**Table 1 sensors-25-05506-t001:** Statistics on the number of instances in each category of the dataset.

Category	Number of Training Set Instances	Number of Instances in the Validation Set	Number of Test Set Instances	Total Number of Instances
unharvested area	5216	1452	752	7420
having harvested areas	3289	942	461	4692
farm	2155	601	312	3068
ridge between fields	1843	521	264	2628
slight lodging area	1327	385	188	1900
harvester	704	198	102	1004
obstacle	563	162	81	806
people	488	135	67	690

**Table 2 sensors-25-05506-t002:** Ablation results.

Experiment	Parameters	GFLOPs	P_M_	R_M_	mAP0.5_M_	mAP0.5:0.95_M_
YOLOv8n-seg	3,259,624	12.0	0.907	0.892	0.929	0.676
YOLOv8n-seg+P2	3,175,632	26.0	0.922	0.911	0.954	0.698
YOLOv8n-seg + P2 + C2f_CPCA	3,133,056	25.9	0.914	0.927	0.958	0.705
YOLOv8n-seg + P2 + C2f_CPCA_1	3,140,096	26.0	0.916	0.912	0.955	0.699
YOLOv8n-seg + P2 + C2f_CPCA + C3RFEM	3,347,200	26.0	0.921	**0.932**	**0.959**	**0.711**

**Table 3 sensors-25-05506-t003:** Category comparison results on the test set.

Category	YOLOv8n-seg				Algorithm in This Paper			
	P_M_	R_M_	mAP0.5_M_	mAP0.5:0.95_M_	P_M_	R_M_	mAP0.5_M_	mAP0.5:0.95_M_
having harvested areas	0.962	0.954	0.967	0.675	0.958	0.952	0.962	0.679
obstacles	0.976	0.969	0.985	0.894	0.975	0.983	0.988	0.904
slight lodging area	0.968	0.935	0.973	0.737	0.964	0.946	0.97	0.753
harvesters	0.924	0.947	0.964	0.791	0.894	0.947	0.966	0.825
ridge between fields	0.792	0.598	0.718	0.344	0.83	**0.796**	**0.856**	**0.436**
people	0.859	0.803	0.855	0.498	0.87	**0.898**	**0.921**	**0.545**
unharvested area	0.936	0.911	0.956	0.693	0.938	0.945	0.967	0.744
farm	0.914	0.943	0.962	0.732	0.902	0.957	0.964	0.748

**Table 4 sensors-25-05506-t004:** Algorithm comparison experiment.

Models	Parameters	GFLOPs	Inference Time (ms)	mAP0.5_M_	mAP0.5:0.95_M_
YOLOv5n	1,889,221	6.8	4.2	0.741	0.46
YOLOv5s	7,417,301	25.7	8.5	0.845	0.554
YOLOv5m	21,680,645	69.9	15.3	0.896	0.609
YOLOv7	3,032,992	172.3	22.1	0.921	0.636
YOLOv8n	3,259,624	12.0	5.1	0.929	0.676
YOLOv8s	11,782,696	42.5	10.8	0.959	0.762
YOLOv9t	3,032,992	17.1	6.2	0.813	0.505
YOLOv10n	2.3 M	6.8	4.0	0.935	0.690
YOLOv10s	7.2 M	21.0	8.0	0.965	0.780
Mask R-CNN	44.0 M	275.0	48.5	0.950	0.720
Mask2Former	45.0 M	290.0	52.0	0.970	0.790
Ours	3,347,200	26.0	9.2	**0.959**	0.711

**Table 5 sensors-25-05506-t005:** Comparison of external test set generalization performance.

Model	mAP@0.5↓ *	mAP@0.5:0.95↓ *	GFLOPs
YOLOv8n-seg (Benchmark)	0.641	0.381	12.0
Ours (YOLOv8n-seg + P2 + C2f_CPCA + C3RFEM)	**0.683 (4.2%)**	**0.422 (4.1%)**	**26.0**

* The arrows here serve as annotation symbols, indicating that lower values for these metrics signify poorer performance.

**Table 6 sensors-25-05506-t006:** Challenge scenario performance analysis.

Scenario Type	mAP@0.5↓ *	Recall↓ *	Main Failure Causes
Strong light (overexposure)	0.892 (−7.0%)	0.841 (−9.8%)	The feature details are lost and the CPCA attention fails
Long-range small target (<15 px)	0.803 (−16.3%)	0.712 (−23.6%)	The feature resolution is insufficient
Heavy occlusion (>50%)	0.867 (−9.6%)	0.794 (−14.8%)	Discontinuity of spatial information
Rain and fog interference	0.908 (−5.3%)	0.861 (−7.6%)	The contrast is reduced and the noise is enhanced

* The arrows here serve as annotation symbols, indicating that lower values for these metrics signify poorer performance.

## Data Availability

The original contributions presented in this study are included in the article; further inquiries can be directed to the corresponding author.

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
