# Peer review of "Enhancing Instance Segmentation in Agriculture: An Optimized YOLOv8 Solution"

_sensors, 2025, doi:10.3390/s25175506_

Round 1

Reviewer 1 Report

Comments and Suggestions for Authors

This paper addresses the segmentation of small and complex objects in complex agricultural scenarios (such as rice field harvesting) by proposing three improvements to the YOLOv8n-seg network (adding a small object detection layer P2, a C2f_CPCA attention module, and a C3RFEM module). The paper validates its effectiveness through ablation and comparative experiments. The overall work demonstrates considerable innovation and engineering application value. Results demonstrate that the improved model significantly improves small object detection and detection of certain categories (such as "field ridge" and "person"), while achieving a good balance between lightweightness and performance. The paper's overall structure is clear, the experiments are comprehensive, and the results are intuitively presented. However, some areas require improvement and further clarification. Specifically:

  1. The abstract needs to be further condensed.
  2. The paper's structure is disorganized; the dataset, evaluation metrics, and experimental environment should not be placed in Chapter 3 (Results).
  3. The paper only provides a comparison of GFLOPs and parameter count, but does not demonstrate inference speed (FPS) or actual deployment results on edge devices. Field testing on agricultural machinery is recommended.
  4. While the paper compares with various YOLO algorithms, it does not include other mainstream segmentation models (such as Mask R-CNN, DETR, and Segment Anything). It is recommended that the authors provide additional comparisons with at least one non-YOLO instance segmentation model to more comprehensively verify the effectiveness of the improvements.
  5. The conclusion section needs to be further condensed, and some of the content can be placed in the discussion section.

Author Response

(1)Response to comment: (The abstract needs to be further condensed.)

Response: Thank you for your suggestions . We have refined and summarised the abstract section more precisely.

(2)Response to comment: (The paper's structure is disorganized; the dataset, evaluation metrics, and experimental environment should not be placed in Chapter 3 (Results).)

Response: Thank you for your suggestions . We also noticed the inconsistency, so we changed the title from ‘Results’ to ‘Experiments and Analysis’ .

(3)Response to comment: (The paper only provides a comparison of GFLOPs and parameter count, but does not demonstrate inference speed (FPS) or actual deployment results on edge devices. Field testing on agricultural machinery is recommended.)

Response: Thank you for your suggestions . We have added inference speed comparison tests to the algorithm comparison in Section 3.4.2.

(4)Response to comment: (While the paper compares with various YOLO algorithms, it does not include other mainstream segmentation models (such as Mask R-CNN, DETR, and Segment Anything). It is recommended that the authors provide additional comparisons with at least one non-YOLO instance segmentation model to more comprehensively verify the effectiveness of the improvements.)

Response: Thank you for your suggestions . We have added several recent models such as YOLOv10n, YOLOv10s, Mask R-CNN, and Mask2Former, which are lightweight models, and conducted a more comprehensive evaluation of the performance advantages of the proposed algorithm.

(5)Response to comment: (The conclusion section needs to be further condensed, and some of the content can be placed in the discussion section.)

Response: Thank you for your suggestions . We have already made more concise statements and condensed the text in the conclusion section .

Reviewer 2 Report

Comments and Suggestions for Authors

The author proposes an enhanced YOLOv8n-seg model for solving instance segmentation tasks in agricultural scenes. By adding a small target detection layer and introducing the C2f_CPCA and C3RFEM modules, the model improves feature extraction capabilities and computational efficiency. Experimental results show improvements in accuracy, recall, and mAP, validating the effectiveness of the proposed enhanced YOLOv8n-seg model. However, the paper has the following issues:

(1)In the introduction, the author summarizes the progress made by existing related studies but lacks a discussion on the shortcomings of these studies. It is recommended that the author clarify the specific problems that this paper aims to solve in relation to the existing research issues.

(2)While the introduction provides the main contributions of the study, these contributions lack specificity and motivation. It is recommended that the author explain in detail how each contribution addresses specific problems found in existing research.

(3)In sections 2.2 and 2.3, the author mentions adding an extra layer specifically for small target detection but does not explain the issues in the original network that lead to poor small target detection performance. The author should further elaborate on the problems the original network encounters when handling small targets and explain why the added layer can compensate for these issues, thereby improving the performance of instance segmentation tasks in agricultural scenes. Additionally, for the sake of academic rigor, in section 2.3, the author mentions: "In segmentation tasks in agricultural scenes, detecting small targets such as people is particularly challenging because these targets are often small in size and irregular in shape, which can easily lead to false detections and poor segmentation." Here, people are referred to as small targets. In agricultural scene instance segmentation tasks, what qualifies as a "large target"?

(4)In section 3.1, the author should provide detailed information about the dataset composition, including the number of samples in different categories such as field intervals, harvested areas, obstacles, lightly fallen areas, harvesters, people, and unharvested areas. Additionally, it is recommended that the author provide specific details about the data collection process, such as the time of collection, lighting conditions, camera parameters, etc. Since the sample size significantly impacts the model's generalization ability, the generalization capability of the proposed enhanced YOLOv8n-seg model may be limited. It is recommended that the author supplement the paper with relevant experiments on model generalization and further discuss the generalization capability based on the experimental results.

(5)In the algorithm comparison experiments, it is recommended that the author include the latest versions of the YOLO models (e.g., YOLOv10, YOLOv11, YOLOv12, etc.) and other network models for comparison, so as to more comprehensively assess the performance advantages of the proposed algorithm.

(6)In the abstract, the author mentions that the proposed model maintains a good balance between computational cost and detection performance. It is recommended that the author include a complexity analysis of all models, including parameter count, computation load (FLOPs), and inference time.

Comments on the Quality of English Language

I recommend professional language editing to enhance clarity and fluency.

Author Response

(1)Response to comment: (In the introduction, the author summarizes the progress made by existing related studies but lacks a discussion on the shortcomings of these studies. It is recommended that the author clarify the specific problems that this paper aims to solve in relation to the existing research issues)

Response: Thank you for your suggestions.We have revised the introduction to discuss the shortcomings of these studies, and provided additional solutions to the existing research problems.

(2)Response to comment: (While the introduction provides the main contributions of the study, these contributions lack specificity and motivation. It is recommended that the author explain in detail how each contribution addresses specific problems found in existing research)

Response: Thank you for your suggestions.We have added specific issues from existing research and provided more detailed solutions in the main contributions based on these issues.

(3)Response to comment: (In sections 2.2 and 2.3, the author mentions adding an extra layer specifically for small target detection but does not explain the issues in the original network that lead to poor small target detection performance. The author should further elaborate on the problems the original network encounters when handling small targets and explain why the added layer can compensate for these issues, thereby improving the performance of instance segmentation tasks in agricultural scenes. Additionally, for the sake of academic rigor, in section 2.3, the author mentions: "In segmentation tasks in agricultural scenes, detecting small targets such as people is particularly challenging because these targets are often small in size and irregular in shape, which can easily lead to false detections and poor segmentation." Here, people are referred to as small targets. In agricultural scene instance segmentation tasks, what qualifies as a "large target"?)

Response: Thank you for your suggestions.We have thoroughly explained the issues with the original network, further elaborated on the problems encountered by the original network when processing small targets, and finally described how the newly added layer compensates for these shortcomings, thereby improving the performance of instance segmentation tasks in agricultural scenarios.It also provides a detailed explanation of the definitions of large and small targets in this study.

(4)Response to comment: (In section 3.1, the author should provide detailed information about the dataset composition, including the number of samples in different categories such as field intervals, harvested areas, obstacles, lightly fallen areas, harvesters, people, and unharvested areas. Additionally, it is recommended that the author provide specific details about the data collection process, such as the time of collection, lighting conditions, camera parameters, etc. Since the sample size significantly impacts the model's generalization ability, the generalization capability of the proposed enhanced YOLOv8n-seg model may be limited. It is recommended that the author supplement the paper with relevant experiments on model generalization and further discuss the generalization capability based on the experimental results)

Response: Thank you for your suggestions. We have provided a detailed description of the composition of the dataset, including the number of samples in different categories and the data collection process. We have deleted Figure 8 and added a table showing the specific sample sizes for each instance. We have also added experiments related to the model's generalization ability and further discussed its generalization performance based on the experimental results.Finally, a more comprehensive analysis is given in Section 4.1.4.

(5)Response to comment: (In the algorithm comparison experiments, it is recommended that the author include the latest versions of the YOLO models (e.g., YOLOv10, YOLOv11, YOLOv12, etc.) and other network models for comparison, so as to more comprehensively assess the performance advantages of the proposed algorithm)

Response: Thank you for your suggestions.We have added several recent models such as YOLOv10n, YOLOv10s, Mask R-CNN, and Mask2Former, which are lightweight models, and conducted a more comprehensive evaluation of the performance advantages of the proposed algorithm.

(6)Response to comment: (In the abstract, the author mentions that the proposed model maintains a good balance between computational cost and detection performance. It is recommended that the author include a complexity analysis of all models, including parameter count, computation load (FLOPs), and inference time)

Response: Thank you for your suggestions.We have already added the inference time of all models, etc., in the algorithm comparison experiment in Section 3.4.2 and conducted a specific analysis.

Reviewer 3 Report

Comments and Suggestions for Authors

1.The Abbreviation should be defined as it appears in the texture at first. For example, C3RFEM, RFEM and so on.

2.Line 186-188, please try to make an explanation or clarification why you want to replace the C2f module within the neck network with the C2f_CPCA module and try to give a reason about that the C3RFEM module is integrated following the SPPF module in the backbone network.

3.Some Figures are not so clear and can not meet the requirements for publication. An improvement is needed.

4.Line 318, what is the C3 module?

5.Line 356-358, that “unharvested area” has the highest proportion, accounting for approximately 40%, ... is not consistence with what the Figure 9 shows.

6.Line 382-385, it is difficult to understand what the figure10-12 want to show.

Comments on the Quality of English Language

No

Author Response

(1)Response to comment: (The Abbreviation should be defined as it appears in the texture at first. For example, C3RFEM, RFEM and so on.)

Response: Thank you for your suggestions. We have unified definitions such as C3RFEM and RFEM.

(2)Response to comment: (Line 186-188, please try to make an explanation or clarification why you want to replace the C2f module within the neck network with the C2f_CPCA module and try to give a reason about that the C3RFEM module is integrated following the SPPF module in the backbone network.)

Response: Thank you for your comments . We have already explained in detail why the C2f module should be replaced with the C2f_CPCA module in the neck network and why the C3RFEM module should be integrated into the SPPF module in the trunk network.

(3)Response to comment: (Some Figures are not so clear and can not meet the requirements for publication. An improvement is needed.)

Response: Thank you for your comments . We have made further updates to image quality .

(4)Response to comment: (Line 318, what is the C3 module?)

Response:Thank you for your comments . We have already explained the meaning of the C3 module in detail in the corresponding section. The C3 module is a key component introduced in the YOLOv5 core network. Its name derives from the combination of the CSP (Cross Stage Partial) architecture and three convolutional layers (Conv). It serves as an efficient, foundational feature extraction unit that enhances gradient flow and information flow.

(5)Response to comment: (Line 356-358, that “unharvested area” has the highest proportion, accounting for approximately 40%, ... is not consistence with what the Figure 9 shows.)

Response: Thank you for your comments . We sincerely apologise for this elementary error. In order to make the data more specific, we have replaced the statistical charts with more specific tabular data.

(6)Response to comment: (Line 382-385, it is difficult to understand what the figure10-12 want to show.)

Response: Thank you for your comments . These figures are tools for scientifically analysing and verifying the effectiveness of ‘data augmentation’ , aiming to visually demonstrate that the data augmentation techniques we employed (contrast adjustment, noise addition, flipping) effectively altered the features of the images . Figure 10 demonstrates that the augmentation operations indeed altered the global brightness statistics of the images , which helps the model adapt to different lighting conditions . Figure 11 demonstrates that the enhancement operations (especially contrast enhancement) significantly impact the edge sharpness of the images . This is critical because instance segmentation models heavily rely on sharp edges to accurately delineate the boundaries between different objects . Figure 12 demonstrates that different enhancement methods indeed create new images with distinct features . The more dispersed and scattered the point cloud distribution , the better the diversity of the enhanced dataset.

Reviewer 4 Report

Comments and Suggestions for Authors

In this paper authors try to address the small target detection problem in complex agricultural scenes by introducing a novel model based on YOLOv8-seg.

Traditional manual methods are inefficient and have limited performance in complex agricultural scenes. The application of deep learning based segmentation methods is limited to instance segmentation, which fails in complex scenes.

The authors based on YOLOv8 model, they introduce novel modules in order to enhance the accuracy of the model in complex agricultural scenes. The main novelties are the small target detection layer, the C2f_CPCA module and the C3RFEM.

  • In Figure 1 authors should describe all blocks that they are using.
  • The authors should add details for the blocks used  in the Figure 3.
  • A short description of SPFF module should also be included.

The conclusions section is consistent and references are appropriate.

Author Response

(1)Response to comment: (In Figure 1 authors should describe all blocks that they are using.)

Response: Thank you for your suggestions . The complete model actually used in this paper is shown in Figure 2 (improved YOLOv8-seg model structure diagram), which includes the improved blocks C3RFEM, C2f_CPCA, and the newly added layer (P2 detection layer). Figure 1 serves only to provide background knowledge.

(2)Response to comment: (The authors should add details for the blocks used in the Figure 3.)

Response: Thank you for your suggestions . We have already described the details of the block in section 2.3 .

(3)Response to comment: (A short description of SPFF module should also be included.)

Response: Thank you for your suggestions . We have briefly introduced the SPFF module when we first mentioned it in Section 2.1 .

(4)Response to comment: (The conclusions section is consistent and references are appropriate.)

Response:Thank you for your comments .

Reviewer 5 Report

Comments and Suggestions for Authors

Author Response

(1)Response to comment: (There is a data reporting inconsistency between the text and Figure 8. The text claims that the obstacle and people categories represent less than 5% of the dataset, but Figure 8 shows these categories with a higher proportion (closer to 10–15%). In addition, the meaning of the “instance” label in the figure should be defined clearly (e.g., whether it represents normalized percentages or absolute counts); otherwise, the figure may mislead readers.)

Response: Thank you for your comments . We sincerely apologise for this elementary error. In order to make the data more specific, we have replaced the statistical charts with more specific tabular data.

(2)Response to comment: (The dataset is imbalanced (e.g., the people category accounts for less than 5%). Providing more details on whether balancing techniques or re-weighting strategies were applied would be valuable for understanding how the model handles class imbalance.)

Response: Thank you for your comments . We also noticed the issue of category imbalance in the dataset. To address this imbalance, we introduced a loss function reweighting method and provided detailed explanations to help understand how the model handles category imbalance issues .

(3)Response to comment: (The dataset is relatively small (593 original images, augmented to 2,372). While augmentation increases the sample size, the limited diversity raises concerns about potential overfitting. A cross-dataset evaluation would help strengthen the generalization claims.)

Response: Thank you for your suggestions . We have therefore introduced Section 3.4.3 , which conducts cross-dataset evaluations to reinforce the claim of generalisation capabilities .

(4)Response to comment: (The article introduces Figures 10–12 as part of the dataset analysis, but the explanations are not sufficiently detailed. The authors should provide a clearer interpretation of what each figure demonstrates and explain how pixel distribution, gradient amplitude, and PCA variation specifically affect the segmentation task and model performance.)

Response: Thank you for your suggestions . We have further explained the contents shown in Figures 10-12 and explained how pixel distribution, gradient magnitude, and PCA variance specifically affect segmentation tasks and model performance .

(5)Response to comment: (The ablation study shows that most of the performance improvement comes from introducing the P2 layer and the C2f CPCA module (YOLOv8n-seg+P2+C2f CPCA). Although the most complex model (YOLOv8nseg+P2+C2f CPCA+C3RFEM) achieves a slight additional performance gain, it also introduces more parameters. The authors should provide a more detailed justification for why the full model is preferable over YOLOv8n-seg+P2+C2f CPCA. Additionally, including this comparison in Table 2 would make the results more convincing)

Response: Thank you for your comments . We have explained in more detail why the complete model is superior to YOLOv8n-seg+P2+C2f CPCA . The current progressive experimental design more clearly demonstrates the incremental contribution of each module and the final combined effect, avoiding the problem of reduced readability caused by too many combinations .

(6)Response to comment: (All figures should be replaced with higher-resolution versions, as some of the text within the figures is blurry and difficult to read.)

Response: Thank you for your comments . We have made further updates to image quality .

(7)Response to comment: (There are question marks in Equations. (4) and (8).)

Response: Thank you for your comments . We have modified these two formulas, which are now formulas 4 and 9 .

(8)Response to comment: (Figures 9(b) and 9(c) has unnecessary labels)

Response: Thank you for your comments . We have deleted unnecessary labels .

(9)Response to comment: (Equations. (7) has a symbol that can not be read.)

Response: Thank you for your comments . We have modified this formula, which is now Formula 8 .

(10)Response to comment: (There is a batch size inconsistency in Section 3.3.)

Response: Thank you for your comments . We are sorry for this mistake and have now standardised the batch .

(11)Response to comment: (. Figure 15 has overlapped legends.)

Response: Thank you for your comments . We have modified the diagram .

(12)Response to comment: (Some sentences need improvement for clarity (e.g., line 318). In addition, there are minor formatting issues , such as missing spaces after words, which should be corrected during proofreading)

Response: Thank you for your suggestions . We have improved the sentences and corrected the formatting issues.

Round 2

Reviewer 1 Report

Comments and Suggestions for Authors The manuscript has beens sufficiently improved. With my best wishes.  

Author Response

Thank you for your comments , we extend our heartfelt best wishes.

Reviewer 2 Report

Comments and Suggestions for Authors

The authors have fully addressed the reviewers’ comments, and the revised manuscript requires no further clarification.

Author Response

(The authors gave the same response as above.)

Reviewer 5 Report

Comments and Suggestions for Authors

The authors have successfully addressed my concerns. However, two minor things need to be improved:

  1. α in Eqn. (5) is not defined.
  2.  Eqn. (8) and (9) still have some symbol issues (see attachment).

Author Response

(1)Response to comment: (α in Eqn. (5) is not defined.)

Response: Thank you for your comments . We have provided a specific definition and explanation for the parameter α.

(2)Response to comment: ( Eqn. (8) and (9) still have some symbol issues (see attachment).)

Response: Thank you for your comments . We sincerely apologise for the garbled text issue and have now rectified the problem.
